# Evaluation of the Versius Robotic System for Infant Surgery—A Study in Piglets of Less than 10 kg Body Weight

**DOI:** 10.3390/children10050831

**Published:** 2023-05-03

**Authors:** Thomas Franz Krebs, Timo Kayser, Ulf Lorenzen, Matthias Grünewald, Marit Kayser, Anna Saltner, Lidya-Olgu Durmaz, Lina Johanna Reese, Ewan Brownlee, Katja Reischig, Jonas Baastrup, Andreas Meinzer, Almut Kalz, Thomas Becker, Robert Bergholz

**Affiliations:** 1Department of General-, Visceral-, Thoracic-, Transplant- and Pediatric Surgery, UKSH University Hospital of Schleswig-Holstein Kiel Campus, Arnold-Heller-Strasse 3, 24105 Kiel, Germany; thomas.krebs@kispisg.ch (T.F.K.);; 2Department of Pediatric Surgery, Ostschweizer Children’s Hospital, Claudiusstrasse 6, 9006 St. Gallen, Switzerland; 3Department of Anesthesia and Intensive Care Medicine, UKSH University Hospital of Schleswig-Holstein Kiel Campus, Arnold-Heller-Strasse 3, 24105 Kiel, Germany; 4Department of Anesthesia and Intensive Care Medicine, Ev. Amalie Sieveking Hospital, Haselkamp 33, 22359 Hamburg, Germany; 5Department of Paediatric Surgery and Urology, Southampton Children’s Hospital, University Hospital Southampton, Southampton SO16 6YD, UK; 6Kurt-Semm-Center for Minimally Invasive and Robotic Surgery, UKSH University Hospital of Schleswig-Holstein Kiel Campus, Arnold-Heller-Strasse 3, 24105 Kiel, Germany

**Keywords:** pediatric surgery, robotics, robotic surgery, minimally invasive surgery

## Abstract

Background: We were able to demonstrate the feasibility of a new robotic system (Versius, CMR Surgical, Cambridge, UK) for procedures in small inanimate cavities. The aim of this consecutive study was to test the Versius^®^ system for its feasibility, performance, and safety of robotic abdominal and thoracic surgery in piglets simulating infants with a body weight lower than 10 kg. Methods: A total of 24 procedures (from explorative laparoscopy to thoracoscopic esophageal repair) were performed in 4 piglets with a mean age of 12 days and a mean body weight of 6.4 (7–7.5) kg. Additional urological procedures were performed after euthanasia of the piglet. The Versius^®^ robotic system was used with 5 mm wristed instruments and a 10 mm 3D 0° or 30° camera. The setup consisted of the master console and three to four separate arms. The performance of the procedure, the size, position, and the distance between the ports, the external and internal collisions, and complications of the procedures were recorded and analyzed. Results: We were able to perform all surgical procedures as planned. We encountered neither surgical nor robot-associated complications in the live model. Whereas all abdominal procedures could be performed successfully under general anesthesia, one piglet was euthanized early before the thoracic interventions, likely due to pulmonary inflammatory response. Technical limitations were based on the size of the camera (10 mm) being too large and the minimal insertion depth of the instruments for calibration of the fulcrum point. Conclusions: Robotic surgery on newborns and infants appears technically feasible with the Versius^®^ system. Software adjustments for fulcrum point calibration need to be implemented by the manufacturer as a result of our study. To further evaluate the Versius^®^ system, prospective trials are needed, comparing it to open and laparoscopic surgery as well as to other robotic systems.

## 1. Introduction 

The development of minimally invasive (MIS) and robot-assisted (RAS) pediatric surgery has led to a reduction of incisional length, risk of infection, postoperative pain, and hospital stay, and improved operative and surgical precision owing to advantages such as magnification, smaller and wristed or angulated instruments, a stable camera view with enhanced 3D vision, and reduction of tremor and downscaling of movements [1,2,3]. Currently, the diameter of the traditional robotic instruments (8 mm or 5 mm with longer articulating ends) and the resulting distance between the ports needed for triangulation of the instruments toward the operative field are limitations for the use of robotic systems in smaller infants [4,5,6,7,8,9,10,11].

The introduction of the Versius^®^ robotic system with its 5 mm instruments and short-wristed ends appears to make interventions on infants and newborns more attainable [9,12,13,14,15]. In an inanimate model, the general application of the Versius^®^ was feasible down to operative spaces of less than 110 mL volume [9].

According to our suggested three-step approach of evaluating new surgical robots with respect to their usability in children, it is necessary to demonstrate the safety and practicability of specific pediatric surgical procedures in a live model. This has not been performed so far. Thus, the aim of the study is to examine the feasibility and safety of pediatric surgical procedures with the Versius^®^ in an animal model simulating infants with a body weight lower than 10 kg [9,10,11,16].

## 2. Materials and Methods

### 2.1. Surgeons

T.F.K. and R.B. were registered as surgeons performing the procedures in this trial. T.F.K. and R.B. are pediatric laparoscopic, robotic, and experimental fetal surgeons. The procedures were performed alternating and acting as a team by interchanging between the roles of operating to assisting the surgeon during the procedures. E.B. performed additional urologic procedures on the euthanized deceased animal.

### 2.2. Animals

This study was approved by the local animal rights and ethics committee (V242–59759/2021, MELUND, Kiel, Germany, 6 October 2021). All animals used in the experimental laboratory were managed in compliance with federal and local laws for animal use and care, according to the ARRIVE guidelines; all institutional and national guidelines for the care and use of laboratory animals were followed [17].

We performed surgery on four piglets with a weight below 10 kg. The animals were kept in isolated groups of three one day prior to the procedure to enable them to adapt to their surroundings, thus reducing stress.

### 2.3. Anesthesia

The anesthesia protocol has been reported before: [11] Premedication of the piglets was initiated intraperitoneally with midazolam (0.5 mg/kg), ketamine (25 mg/kg), and atropine (0.04 mg/kg). Anesthesia was induced with a propofol bolus dose (1–3 mg/kg intravenously) and ketamine (1–2 mg/kg) and maintained with a continuous infusion of propofol (4–6 mg/kg/h) and ketamine (0.05–0.15 mg/kg/h) via an ear vein. After endotracheal intubation during spontaneous respiration, the animals were ventilated pressure-controlled with 40% oxygen and 25–35 breaths/min and with a tidal volume of 10 mL/kg. The inspiratory time (Tinsp) was 1.0 s (0.7–2.0 s), the positive endexspiratory pressure (PEEP) was adjusted to 3–5 cm H_2_O and the inspiratory pressure (Pinsp) was adjusted up to 20 cm H_2_O. Ventilation was performed with the Draeger Primus (Dräger, Germany) and monitored oxygen and end-tidal carbon dioxide. Oxygen saturation (SpO_2_) was monitored by a continuous pulse oximeter (DATEX OHMEDA S/5, Compact-Monitor) placed on the animal’s tail. Meloxicam was applied intra-muscularly (0.4 mg/kg) every four hours for analgesia. Rocuronium (0.5 mg/kg) was given as needed. The depth of anesthesia was judged based on heart rate and respirations as well as reaction to stimuli. If clinical assessment suggested a decreasing level of anesthesia, additional propofol, midazolam, and ketamine were injected. Body temperature was monitored by a rectal temperature probe and maintained between 37 and 38 °C with a heating blanket.

At the end of the procedures, euthanasia was performed under anesthesia by intravenous administration of potassium chloride (20 mmol). The animals were observed for 20 min to verify they were deceased.

### 2.4. Robotic System and Instruments

Two Versius^®^ robotic systems with 5 mm instruments, six instrument/camera arms, and monopolar and bipolar energy devices were supplied by CMR as a research grant for R.B.

The setup (Figure 1) consisted of three to four separate arms, each on its own mobile base with an average weight of 100 kg. The camera (a 10 mm 3D with either a 0° or 30° scope) was attached to a separate arm. The master console consists of an HD 3D Monitor used with passive 3D glasses. The console grips are a mix of forceps-like handpieces for the index finger and thumb and pistol grips. The instruments, consisting of passive, bipolar, or monopolar graspers, scissors, and hooks, were applied as judged by the operating surgeons. The system was covered in sterile drapes and connected to a video system with an additional external monitor and recording capabilities (Sony, Berlin, Germany). CO_2_-Insufflation was applied with the Endoflator^®^ (Karl Storz, Tuttlingen, Germany), and suction and irrigation were generated with the Unimat30^®^ (Karl Storz, Tuttlingen, Germany). The sutures used were documented per procedure. All sutures were tied with intracorporal knots, as we deemed the robotic system to be especially beneficial in laparoscopic intracorporal knot tying.

### 2.5. Surgical Procedures

The setup, the procedures, and their evaluation were adapted from an earlier study of our group: [11] All procedures were performed under general anesthesia in intubated and ventilated animals. The animals were placed in the supine or prone position, and the abdomen or thorax was washed, shaved, and disinfected. The robotic arms were set according to the anticipated procedure, and the sterile field and robot were draped.

The first procedure evaluated was the calibration of the system for calculating the fulcrum point: The Versius^®^ system measures forces at the instrument shaft and calculates the fulcrum point causing the lowest strain to the abdominal wall. For this calculation, the instrument needs to be inserted at least 2 cm into the body cavity. This distance was adapted from originally 5 cm upon results from our study in inanimate models [9]. The results of these measurements depend on the port the instrument is inserted through, the insertion depth of the port and instrument, their rigidity as well as tissue elasticity. It can be hypothesized that in newborns with much higher compliance of the abdominal wall, the calculation of the fulcrum point may fail because of the lack of sufficient force feedback from the abdominal wall, not concluding the appropriate fulcrum point or even damaging the child. The fulcrum point calibration was performed with all used instruments on every arm of the system. At the start of the study, a specific setup with a three-port approach with different insufflation pressures ranging from 2 to 16 mm of mercury and applying different instruments and cameras was used to evaluate the fulcrum point calibration for laparoscopy and thoracoscopy in the animals. Any collision or bending of the instruments as well as excessive depression with deformation of the body wall was recorded.

The procedures selected for evaluation were considered to be the most common minimally invasive pediatric procedures. In case of unforeseen complications, the live procedure was terminated and completed in the sacrificed animal. Conversion to open or laparoscopic surgery for completion of the procedures was not implemented.

The procedures consisted of fulcrum point calibration, explorative laparoscopy, abdominal wall reconstruction, suture ligation of the umbilical vein, vesicocutaneostomy, nephroureterectomy and pyeloplasty, entero-enterostomies, gastric wedge resection, atypical liver wedge resection, splenectomy, diaphragmatic plication, and closure of a diaphragmatic hernia, Nissen fundoplication and hiatoplasty, cholecystectomy, esophageal resection and anastomosis, cholecystoenterostomy, and lobectomy of the right upper lobe. 

### 2.6. Evaluation of the Surgical Procedures

The evaluation of the surgical procedures was standardized according to a previously reported protocol [11]. The amount, size, and position of the applied ports, their distance (in cm, ΔLC: distance of the left-hand instrument to the camera, ΔRC: distance of the right-hand instrument to camera and ΔLR: distance of the left-hand to right-hand instruments, respectively, ΔALC: distance between auxiliary left hand to camera port, ΔARC distance between auxiliary right hand to camera port) were recorded [10]. The abdominal dimensions of the piglets were recorded as abdominal length (LENGTH: distance between the xiphoid process and the pubic tubercle in centimeters) and abdominal width (WIDTH: maximum distance between the left and right abdominal wall in the supine position).

All procedures were video-recorded for later blinded analysis. Outcome parameters were set as follows: completion of the task (yes, no), number of relevant external or internal instrument–instrument collisions (n), and instrument–organ collisions (n).

Rational data were given as mean and range due to the pilot character of this study, and no comparison with other studies or statistical analyses were performed. We discussed technical issues and the progress of the study on a daily basis with the accompanying CMR staff.

## 3. Results

Four piglets were used; see Table 1 for abdominal dimensions and distances between the operative ports. Relevant external or internal instrument–instrument collisions were detected with a median rate of two per procedure. All could be resolved by repositioning the robotic arms. No relevant instrument–organ collisions occurred.

### 3.1. Fulcrum Point Calibration for Laparoscopy and Thoracoscopy

Calibration for the fulcrum point was successful in most of the procedures; 0.5% of cases needed two attempts. Bending of the instruments was not encountered, neither during calibration of the fulcrum point nor during the procedures [11].

### 3.2. Explorative Laparoscopy

Inspection and evaluation of the four abdominal quadrants were performed by a three-port approach with two instruments. We managed to accomplish handling and running the bowel. It was possible to change the workspace from one quadrant to that which was horizontally adjacent without manually repositioning the ports. By contrast, changing workspaces and quadrants in a vertical manner required manual repositioning of the camera and the robotic arms.

### 3.3. Abdominal Wall Reconstruction

After explorative laparoscopy, the abdominal wall was incised in the left upper quadrant, and the musculature and peritoneum were then sutured by interrupted and running sutures (Vicryl 3-0 RB-1, Ethicon, Johnson and Johnson, Hamburg, Germany) simulating abdominal wall hernia repairs.

### 3.4. Suture Ligation of the Umbilical Vein

The piglet was placed in the anti-Trendelenburg position, the camera port was placed 3 cm below the umbilicus, and the left and right robotic arms were placed in triangulation to the umbilical vein.

The umbilical vein was fixed with a grasper in the left arm and dissected from the abdominal wall and toward its entry into the liver. Upon complete mobilization, it was suture ligated on both ends (Vicryl 5-0 TF-1, Ethicon, Johnson and Johnson, Hamburg, Germany) and then transected. The resected part was extracted from the abdomen through the right-hand port.

### 3.5. Vesicocutaneostomy

The piglet was placed in the Trendelenburg position, the camera port was placed in the umbilicus, and the left and right 3 mm robotic arm was in triangulation to the dome of the bladder. The dome of the bladder was identified and opened with monopolar hook cautery. The opening was sutured to a corresponding incision in the abdominal wall with interrupted 10 cm 3-0 Vicryl SH sutures. Then, the sutures of the cystocutaneostomy were taken down, and the defect in the bladder was closed with interrupted 10 cm 3-0 Vicryl SH sutures.

### 3.6. Nephroureterectomy (One (Left) in the Live Animal and Two (Each One Right and Left) in the Euthanized Animal)

The piglet was placed in an almost prone to 20° dorsally rotated position. Access to the right kidney was gained with the camera port, which was inserted 2 cm to the right of the umbilicus, and the left and right arms were positioned in triangulation to the right kidney. An additional port for retraction of the bowel was inserted caudally into the left arm port.

After visualization of the right kidney, the afferent and efferent vessels and the ureter were isolated. The vessels were separated under 5-0 Vicryl ligatures. After vascular control, the kidney was mobilized, the ureter suture ligated (5-0 Vicryl), and cut next to its entry into the bladder. The kidney was not extracted through an incision to the outside of the abdominal cavity but placed into the pelvis, as multiple procedures were performed in those piglets.

In the euthanized animal, the same setup and approach were used as for the pyeloplasties below: Initial dissection was performed with a fenestrated atraumatic grasper in the left hand and bipolar forceps on the right. The renal hilar vessels were each identified, dissected, controlled, and cauterized using the bipolar forceps, then divided using the curved scissor instrument. Attention was then turned to the rest of the kidney, which was dissected out, again using the fenestrated grasper in the left hand and bipolar forceps in the right, starting working cranially and then caudally until fully free. The ureter was then divided using curved scissors, ready for the specimen to be removed.

### 3.7. Pyeloplasty (One on a Left Kidney in a Live Animal and One Left- and One Right-Sided Pyeloplasty in a Euthanized Animal Each)

For the procedure in the live animal, the piglet was placed in an almost prone to 20° dorsally rotated position. Access to the left kidney was gained with the camera port inserted 2 cm to the right of the umbilicus, and the left and right arms were positioned in triangulation to the left kidney. An additional port for retraction of the bowel was inserted caudally into the left arm port. The kidney was exposed, and the vessels were identified and marked with loops (Vicryl 3-0). The renal pelvis was isolated and cut longitudinally. Transverse reconstruction was performed with interrupted 5-0 Vicryl TF-1 sutures.

For the procedures in the euthanized animal, in a lateral position with the ipsilateral side up, three modular robotic bases (camera and two operating modules) were positioned on the contralateral side of the operating table. Following insertion of the umbilical port using standard cutdown technique, the location of the kidney was identified to then allow careful triangulation of two 5 mm ports, which were inserted under vision. No additional ports were required. Initial dissection was performed with a fenestrated atraumatic grasper in the left hand and bipolar forceps in the right. Once the pyeloureteric junction was carefully delineated, in the absence of intrinsic obstruction, an incision was made extending from the mid-pelvis a considerable way down the ureter using a curved scissor instrument to simulate the normal situation of the ureter. An anastomosis was then completed with 5-0 Vicryl in one case and 6-0 PDS in the other, using two needle holders.

### 3.8. Entero-Enterostomies

The camera port was placed in the umbilicus, and the left and right arm ports were in triangulation to the gastric greater curvature. Two loops of the small intestine were placed next to each other, incised longitudinally with monopolar hook cautery, and anastomosed with an interrupted or running suture each (5-0 Vicryl TF-1, 10 cm).

### 3.9. Gastric Wedge Resection

The camera port was placed in the umbilicus, and the left and right arm ports were in triangulation to the gastric greater curvature. The greater curvature was dissected, and a wedge was resected, applying a laparoscopic stapler (Just Right, Hologic, Marlborough, MA, USA) via an assistant port.

### 3.10. Atypical Liver Wedge Resection

The camera port was placed in the umbilicus, and the left and right arms were in triangulation to the left lobes of the liver. The piglets’ livers were extremely vulnerable to grasping and retraction. The most peripheral lobe was visualized, and an atypical wedge resection of a representative part of the liver was resected with monopolar and bipolar hemostasis. We did not encounter any bleeding or bile leakage during follow-up while operating on other abdominal organs.

### 3.11. Splenectomy

The camera port was placed in the umbilicus, and the left and right arms were in triangulation to the spleen. The surrounding tissue was dissected from the splenic hilum, and the hilar vessels were transected, applying a laparoscopic stapler (Just Right, Hologic, Marlborough, MA, USA) via an assistant port.

### 3.12. Diaphragmatic Plication and Closure of a Diaphragmatic Hernia

The piglet was placed in anti-Trendelenburg’s position. The liver was retracted by a 5 mm blunt grasper inserted through an additional port in the left upper quadrant. The camera port was placed in the umbilicus. The left- and right-hand ports were placed in triangulation to the esophagus. The left diaphragm was exposed and incised horizontally in the lumbocostal region to simulate a Bochdalek diaphragmatic defect. The defect was then closed with interrupted sutures (Ethibond 2-0). Over the closed defect, the diaphragm was plicated in a vertical mattress fashion with interrupted 2-0 Ethibond slipping knots. A similar approach was performed on the right side (Figure 2).

### 3.13. Nissen Fundoplication and Hiatoplasty

The piglet was placed in an anti-Trendelenburg position. The liver was retracted by a third robotic instrument inserted through an additional port in the left upper quadrant. The camera port was replaced from the umbilicus to a position 3 cm in the midline above the umbilicus for better access to the subdiaphragmatic region, and the left- and right-hand ports were placed in triangulation to the esophagus.

After visualization and dissection of the esophagus, the fundus was mobilized with bipolar cautery. Access to the hiatus was obstructed by the vulnerable liver. Careful retraction and dissection exposed the hiatus, which resulted in the opening of the right hemithroax. Hiatoplasty was performed by interrupted 3-0 Vicryl sutures. A gastric 360° Nissen wrap was placed around the esophagus and stitched with interrupted sutures (3-0 Vicryl S-H, Ethibond, Norderstedt, Germany).

### 3.14. Cholecystectomy

The piglet was placed into anti-Trendelenburg’s position and turned slightly to its left side. The three robotic ports were placed with triangulation into the area of the gallbladder with the camera port in the umbilicus. The liver was elevated by applying a third robotic instrument port and a long grasper. The neonatal piglet liver, as well as the gallbladder, is very sensitive to mechanical stress, as reported earlier [11]. After elevation of the liver and retraction of the gallbladder, the infundibulum, the cystic duct, and the cystic artery were identified. Ligation of the cystic artery and duct was accomplished with 5-0 Vicryl RB-1 (Ethicon, Norderstedt, Germany). Gallbladder dissection from the liver was performed with monopolar hook cautery.

### 3.15. Esophageal Resection and Anastomosis

Access to the right thorax was gained with the lung compressed by insufflation of CO_2_ with a pressure starting with 4 mmHg. The piglet was ventilated bilaterally and placed almost prone. The camera port was placed about 2 cm para-spinally to the right in an extension of the right eye in the mid-thoracic height. The ports for the left and right instruments were placed cranially and caudally to the camera port and more anterior situated. Access to the porcine thorax was possible, although the narrow intercostal spaces hampered manipulation, especially with respect to the 10 mm camera.

The esophagus was isolated while sparing the vagal nerve. An approximately 8 mm long segment was resected, and the ends anastomosed end to end with interrupted Vicryl 5-0 TF-1 suture (Figure 3). 

### 3.16. Cholecystoenterostomy

The piglet was placed into the anti-Trendelenburg position and turned to its left side. The three robotic ports were placed with triangulation into the direction of the gallbladder with the camera port in the umbilicus. Due to the anatomy of the porcine liver, a fourth port was needed in the left upper abdomen for robotic liver retraction. A 2 mm incision was set in the fundus of the gallbladder with monopolar hook cautery. An adjacent loop of the small bowel was opened longitudinally with cautery to a corresponding length. The omega-shaped anastomosis was performed with interrupted sutures, the dorsal knots pointing to the inside and the ventral ones to the outside (6-0 Vicryl TF-1, Ethicon, Norderstedt, Germany, Figure 4).

### 3.17. Lobectomy of the Right Upper Lobe

The piglet was placed in a near-prone position, and the camera port was placed about 5 cm paraspinally to the right at mid-thoracic height. The ports for the left and right instruments were placed cranially and caudally to the camera port and more posteriorly situated for triangulation to the lung. For retraction, an additional port was placed caudally to the left arm port.

The thorax was insufflated with CO_2,_ and the piglet was ventilated in both lungs. The right upper lobe, the vessels, and the bronchus were identified. The bronchus was cut between ligatures, and the vessels were divided between ligatures (5-0 Vicryl TF-1). The upper lobe was then completely mobilized but not extracted.

### 3.18. Complications and Technical Limitations

We did not encounter any surgical or robot-associated complications during the study. The minimum insertion depth of at least 2 cm, which was required for successful fulcrum point calculation, did not lead to any complications. One animal had to be euthanized due to intractable pneumonitis, which aggravated before the first planned thoracic procedures could be commenced: However, we decided to conduct the procedures on this animal despite the circumstances. Thankfully, all subsequent animals that underwent the thoracic procedures remained alive, and we did not encounter any morbidity or mortality in them.

## 4. Discussion

We were able to successfully perform all planned pediatric surgical procedures in piglets with a mean body weight of fewer than seven kilograms. Based on our previous experience with Nissen fundoplication in piglets applying the Senhance, we did not encounter any complications during this study [11]. This might be related to the articulating wristed instruments that may improve dexterity in confined spaces.

### 4.1. Limitations

Our report is a non-comparative evaluation of the general feasibility, safety, and technical limitations of a specific robotic system (Versius^®^, CMR, Essex, UK) in piglets of fewer than 7 kg, thus simulating neonatal and infant robotic-assisted surgery. We did not compare this system to open, laparoscopic or robotic procedures with the da Vinci^®^ or Senhance^®^ system. Therefore, no definite conclusions should be drawn concerning any inferiority or superiority of this system to existing pediatric surgical techniques.

The live surgical procedures were performed by dedicated pediatric surgeons (T.F.K., R.B.) under experimental settings with no pressure of time. Any conclusions regarding the wider application of the robotic system with pediatric cases in clinical settings, where time limitation and the need to achieve optimal results are crucial factors, have to be drawn with extreme caution.

The aim was not to compare specific surgical techniques; therefore, we did not record any scores like the Objective Structured Assessment of Technical Skills (OSATS) [18,19,20]. After the demonstration of the general safety and feasibility, the next step should be a direct comparison of Versius-based procedures with the corresponding open, laparoscopic, and robot-assisted procedures in simulated infants. In such a setting, the recording and evaluation of specific scores will be a useful instrument to compare the respective performance.

Simulation of infant pelvic procedures is crucial for pediatric robotic surgery, especially in cases such as Hirschsprung’s disease, anorectal malformations, and cloaca. However, it is worth noting that piglet pelvic anatomy differs somewhat from that of human infants, with a narrower, more V-shaped pelvis and a mobile bladder located pseudo-intraperitoneally. In this study, we performed rectal dissections down to the pelvic floor and mobilized the uterus and bladder, but we did not evaluate specific procedures and thus did not include them in our report.

Furthermore, robotic procedures should also be evaluated in animal models of less than 5 kg down to 1500 g or less. We are currently working on a neonatal piglet model for esophageal surgery. Once established, robotic procedures can be evaluated with this model, too. Up to now, we have demonstrated the feasibility of robotic procedures with the Versius system in animals with a mean body weight of less than 7 kg, which simulates human infants between one and three months of age.

### 4.2. Complications

Although we did not experience any surgical or robot-associated complications, one animal had to be euthanized due to pneumonia, which occurred under anesthesia after the completion of the abdominal procedures and prior to the thoracic interventions. One may argue that this piglet should have been excluded from the study, but veterinary examinations just before induction of the anesthesia did not reveal any underlying illness for the discontinuation of the procedures.

### 4.3. Fulcrum Point Calculation

The Versius^®^ system requires the instruments to be inserted at least 2 cm inside of the abdominal cavity for successful calculation of the fulcrum point. This minimum insertion depth was reduced from 5 to 2 cm based on our evaluation in inanimate models [9]. In our experimental studies, the minimal depth of 2 cm did not interfere with the calculation of the fulcrum point or lead to any visible excessive force, evaluated by inspection during the calculation, to the abdominal wall. In contrast to the Senhance^®^ robotic system, the Versius^®^ identifies its fulcrum point by the surgeon moving the instrument in a clockwise direction and not by the robot pressing it automatically downwards until the force is centered on the instrument as is the case with the Senhance. The risk of causing injury to a young patient by an instrument that moves automatically thus appears much smaller.

The refinement of movements through the recalibration of the fulcrum point was found to be unnecessary during our procedures. The system’s instrument memory feature allowed for multiple instrument exchanges without requiring recalibration, thereby avoiding any prolongation of the workflow. Interestingly, we discovered that calibration of the fulcrum point could also be performed ‘in free air’ by manually holding the instrument at the desired fulcrum point during calibration, providing a workaround for repeated unsuccessful calibration attempts. While we applied this method in our study, it was not found to be necessary for docking the system. However, it does afford the surgeon greater flexibility in determining the optimal placement of the fulcrum point.

### 4.4. Collisions

Instrument–organ and instrument–instrument collisions appear to increase with decreasing surgical space [10,11]. The risk of involuntary collisions with potentially life-threatening complications is thus a not to be underestimated risk in small children. As this study was conducted to test the principal feasibility and safety of robotic procedures in simulated infants, we cannot draw any conclusions on the optimal placement of the three robotic manipulator arms. These data have to be obtained by a trial comparing specific procedures performed by open, laparoscopic, and robotic surgery.

Nevertheless, careful positioning of ports is essential; this is all the more crucial, as the smaller the operating space becomes, the closer the ports are to each other and the closer the operating target is to the fulcrum point of the port. In our cases, the use of the 30-degree scope becomes even more useful to move the camera arm away from the instruments; as one moves more instruments toward the edge of the screen, the external arms move closer together; therefore, it appears to be of utmost importance to keep the instruments close to the center of the screen in these small patients.

### 4.5. Advantages and Disadvantages of the Versius^®^ Compared to the Senhance^®^ and to the da Vinci^®^ System

We have experienced the following advantages and disadvantages based on our studies and clinical applications: the Versius^®^ offers 5 mm diameter wristed instruments, which have shorter angulated ends than the corresponding da Vinci^®^ instruments which appears to be advantageous in small spaces, although a comparative study of both systems is still lacking. The main advantage of the Senhance is that the 3 mm diameter instruments enable downscaling of surgical space with a more probable application in newborns and infants, but the instruments are not wristed, therefore providing no improved manual dexterity compared to traditional 3 mm instrument laparoscopy except tremor filtering and movement scaling [9,11,16]. All systems offer 3D stereoscopic vision, although the closed console of the da Vinci^®^ appears to improve the “immersion” of the surgeon into the operative field, similar to microscopy, as compared to the open console with stereoscopic glasses of the Versius^®^ and Senhance^®^. 

Tactile force feedback is offered only by the Senhance^®^ and may improve safety compared to the da Vinci^®^ and Versius^®^.

Considering the multitude of surgical robotic systems available, numerous platforms share common abovementioned functionalities. Among these features, the Versius^®^ system, along with the Senhance^®^ system, enables instruments to be directly inserted into the patient without any ports. This is possible due to the automated calculation of the fulcrum or pivot point by the system, resulting in enhanced flexibility compared to robotic systems that require port docking for surgical access.

Comprehensive data on infant robotic surgery is rare [14,15]. There is no current data on robotic surgery in infants with the Versius^®^ and not much data concerning infant surgery with the da Vinci^®^ or Senhance^®^ system [21,22].

Whether a general recommendation should be given for robotic infant surgery needs to be discussed based on the available data. We have proposed a four-step approach to evaluating the usability of robotic surgical systems in infant procedures (1. inanimate model, 2. live animal model, 3. comparison of open, laparoscopic, and robotic procedures in live animal models, and 4. human infant application) [11].

In contrast, one may find it would be rather unethical to consider comparative studies in animals. Using the IDEAL collaborative process, stage IIa for the first human studies could also be considered, as the next stage will be to look at safety and efficacy [23,24].

Any application of robotic systems in neonates or infants has to be proven equivalent to being safe and effective in open or laparoscopic procedures in comparative animal studies before applying this device in newborns without reliable data. This is of utmost importance, especially when considering that in early-stage uterine cervical carcinoma, randomized prospective data have demonstrated a negative effect of minimally invasive laparoscopic or robotic surgery on patient survival compared to open surgery [25,26,27].

The emergence of digital surgery and the Includenternet of Surgical Things, includeding robotic-assisted surgery, has paved the way for a more collaborative approach to surgery, enabling long-distance telesurgery and telementoring. The integration of interconnected robotic-assisted surgical devices, augmented reality, artificial intelligence, image-guided surgery, and autonomous interventions promise to enhance surgical workflow in the operating theater and ultimately bolster patient safety [28].

## 5. Conclusions

Robotic newborn and infant surgery appear technically feasible with the Versius^®^ robotic system. We have been able to successfully complete 24 procedures without any surgical or robot-associated complications.

Further evaluation of this system is needed by prospective experimental and later clinical trials to compare it to open, laparoscopic, and other robotic procedures.

## Figures and Tables

**Figure 1 children-10-00831-f001:**
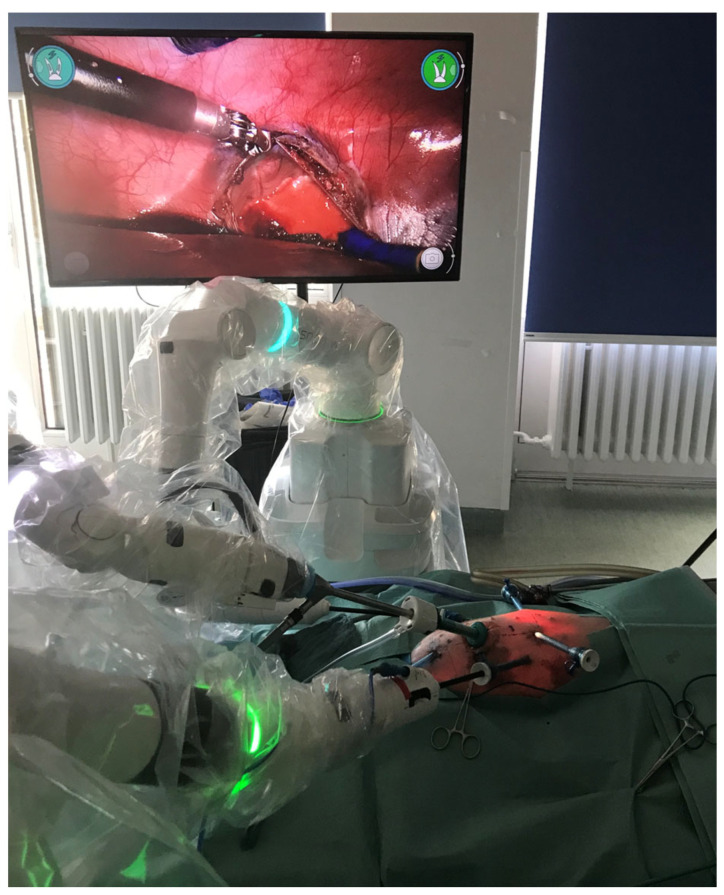
Setup of the experiments. The Versius can be seen with its three arms, operating a 10 mm camera and two 5 mm instruments in the right- and left-hand arms. Three assistant ports for previous procedures can be seen. An accessory video screen for the 2D display of the operating field is on the piglet’s right side.

**Figure 2 children-10-00831-f002:**
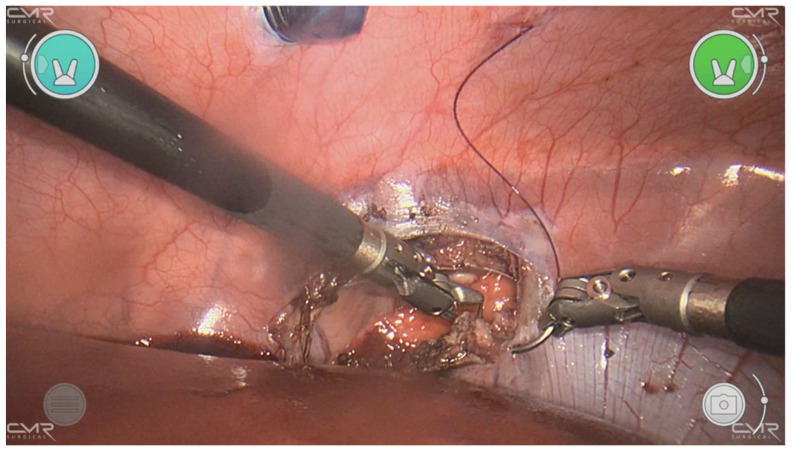
Transabdominal closure of a right-sided diaphragmatic defect.

**Figure 3 children-10-00831-f003:**
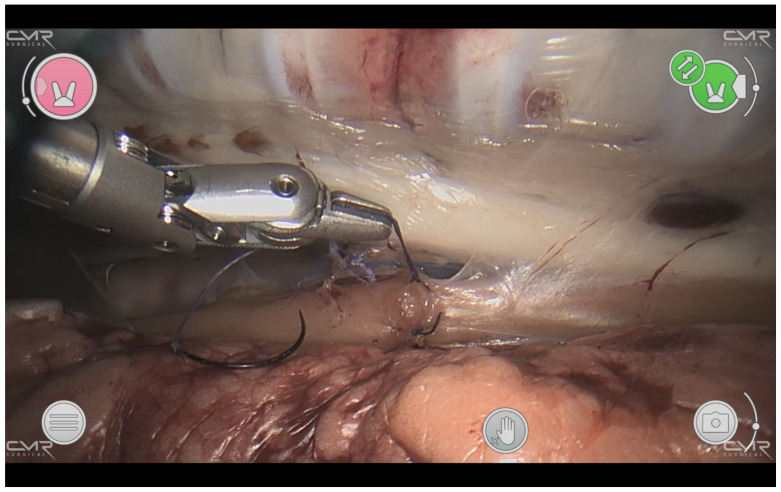
Thoracoscopic esophageal anastomosis. The esophagus with interrupted stitches can be seen horizontally. A 5-0 Vicryl TF-1 suture is in the surgeon’s left instrument, which has a diameter of 5 mm and demonstrates the confined space that has to be worked in.

**Figure 4 children-10-00831-f004:**
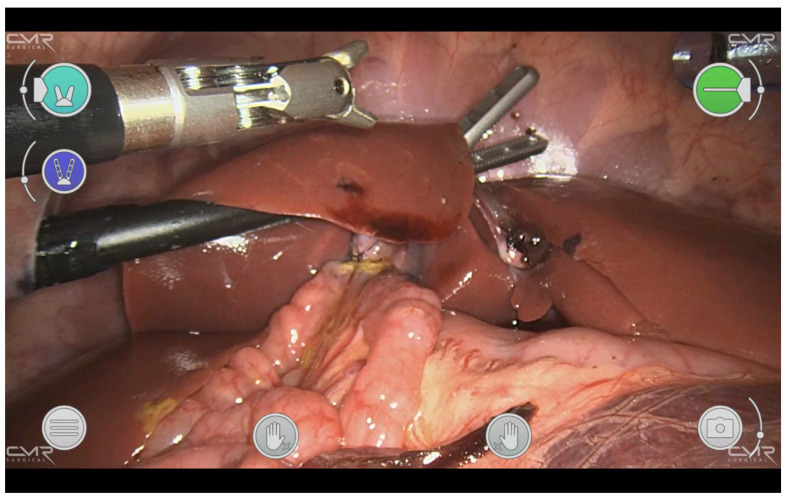
The view is directed onto the omega-loop cholecysto-enterostomy, which was sutured with Vicryl 6-0 by a four-arm approach with two left-side and right-sided robotic 5 mm instruments.

**Table 1 children-10-00831-t001:** Animals used and procedures performed per animal. ΔLC: distance of the left instrument to camera; ΔRC distance of the right instrument to camera; ΔLR distance of the left instrument to the right instrument; * at the height of the umbilicus, all data given as median of all procedures performed per piglet.

ΔLR (cm)	ΔLC (cm)	ΔRC (cm)	Abdominal Diameter (cm) *	Puboxiphoid Length (cm)	Weight (kg)	Piglet No
15	7	8	16	29	7	1
15	7	8	16	26	7.5	2
14	7.5	7	15	25	6	3
10	4	6	14	24.5	5	4
11	5	6	thoracic procedures			2,3,4
*13*	*6.1*	*7*	*15.25*	*26.125*	*6.375*	*MEAN*

## Data Availability

The datasets generated and/or analyzed during the current study are not publicly available but are available from the corresponding author upon reasonable request.

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
