# Peer review of "Evaluation of the Versius Robotic System for Infant Surgery—A Study in Piglets of Less than 10 kg Body Weight"

_children, 2023, doi:10.3390/children10050831_

Round 1

Reviewer 1 Report

It’s such a great privilege to review this manuscript it addresses an important topic which is use if robotic surgery in neonates, many surgeons who are even pioneers in MIS paediatric surgery have concerns about its application in neonates, yet if we thought about the technological advancement put into robots and it’s potential for autonomy or even  a moderate degree of autonomy it could be rendered safe. An area which could be improved and becomes autonomous at least when it comes to port and pressure adjustments such (automatic adjustment and feed back of inappropriate pressure applied to skin surface) would be pivot point adjustment the fact that 2 cm introduction from 5 cm that is an improvement next step hopefully this would be eliminated.

The decrease in instrument size is also an corner stone in establishment of robotic system in neonates and I am glad versius caught up from where zeuss stopped.

I have just few recommendations in neonates the weight is as low as 2 kg and 1.5 kg when we operate on them, it would be worth trying to operate on such weight as I see cut off point in your study is 7 kg.

The robotic system in the thoracic procedure in your study was carried out on an euthanised animal, of all the procedures thoracic procedures are most important to carry out when animal is alive and breathing because in real life anathesia can be challenging and lung movements has significant impact on instrument movement and visualisation and automatic adjustments of insufflation pressures as well please  cite

Alaa Obeida, Basma Magdy, Hebatallah Taher, Mohamad Qinawy, Mohamed Elbarbary,

Absent Azygos Vein in a neonate with oesophageal atresia and tracheoesophageal fistula,

Journal of Pediatric Surgery Case Reports,

Volume 90,

2023,

102588,

ISSN 2213-5766,

https://doi.org/10.1016/j.epsc.2023.102588.

One other important procedure or area which was not mentioned was pelvic operation, given the weight of the piglets it’s ideal of infants in first 6 months of life the time during which we do carry out several pelvic procedures.

The pelvic hiatus is very challenging to access open approachand MIS is proven to be better however  in such narrow spaces ergonomics can be challenging especially in those children whose pelvis is anatomically distorted and become smaller than usual such as patient with bladder exstrophy and cloaca patients please cite 

Taher HMA, Fares A, Wishahy AMK. Laparoscopic Resurrection of an Old Technique: A New Approach for Total Urogenital Separation and Rectal Pull-Through in Patients with Long Common Channel Cloacal Malformation. J Endourol. 2022 Sep;36(9):1177-1182. doi: 10.1089/end.2021.0724. Epub 2022 Mar 7. PMID: 35152733; PMCID: PMC9422784.

Taher H, Elboraie A, Fares A, Tawfiq S, Elbarbary M, Abdullateef KS. Laparoscopic inguinal hernia repair in bladder exstrophy, a new modified solution to an old problem: A cohort study. Int J Surg Case Rep. 2022 Jun;95:107252. doi: 10.1016/j.ijscr.2022.107252. Epub 2022 May 25. PMID: 35636219; PMCID: PMC9157442.

With remarkable advances of computer vision haptic feedback is diminishing in significance, 3D stereotypic vision is available in your new system which is good what kind of computer vision options are available and can be potentially available.

I would suggest limitations of the study or suggestions for future improvements would be (rather than comparative studies) operating on 2 kg animals, thoracicscopic procedure on a live animal once this is carried out and proven successful the next step would be comparative studies.

Author Response

and  here:

It’s such a great privilege to review this manuscript it addresses an important topic which is use if robotic surgery in neonates, many surgeons who are even pioneers in MIS paediatric surgery have concerns about its application in neonates, yet if we thought about the technological advancement put into robots and it’s potential for autonomy or even a moderate degree of autonomy it could be rendered safe.

Thank you very much for your supportive comments.

An area which could be improved and becomes autonomous at least when it comes to port and pressure adjustments such (automatic adjustment and feed back of inappropriate pressure applied to skin surface) would be pivot point adjustment the fact that 2 cm introduction from 5 cm that is an improvement next step hopefully this would be eliminated.

Thank you very much, you are indeed raising a very important topic: CMR has addressed the issue of insertional depth of the instruments for pivot point calibration after results from our previous studies in inanimate models as reported in this manuscript. Nevertheless, an automatic measurement of forces at the instrument and the point of entrance into the body of the child with automated pressure and force adjustments by AI is absolutely desirable

The decrease in instrument size is also an corner stone in establishment of robotic system in neonates and I am glad versius caught up from where zeuss stopped.

Thank you very much for this comment and we totally agree with you that a reduction of the diameter of the instruments while maintaining the wristed aspect and the degrees of freedom is absolutely desirable for pediatric infant surgery. Although procedures have been technically possible in the simulated infants with the 5mm and 6 mm instruments, having wristed 3 mm robotic instruments would be a huge step forward for all pediatric surgeons.

I have just few recommendations in neonates the weight is as low as 2 kg and 1.5 kg when we operate on them, it would be worth trying to operate on such weight as I see cut off point in your study is 7 kg.

Thank you for your comment. We agree that performing laparoscopic procedures on neonatal and preterm babies, who often weigh as little as 1.5 kg or less, is desirable. However, our study aimed to evaluate robotic assisted surgery in animals weighing less than 10 kg, which is the cut-off for CE certification of robotic systems such as the Senhance system. While we are working on developing an animal model using newborn piglets weighing 1 to 1.5 kg for further studies, we opted to use piglets weighing around 5 kg for this study. This weight provided a balance between stability during surgery and the difficulties associated with anesthesia for abdominal and thoracic procedures in newborn piglets.

The robotic system in the thoracic procedure in your study was carried out on an euthanised animal, of all the procedures thoracic procedures are most important to carry out when animal is alive and breathing because in real life anathesia can be challenging and lung movements has significant impact on instrument movement and visualisation and automatic adjustments of insufflation pressures.

Thank you for your comment, and we completely agree with your point. While planning for thoracic procedures in our study, we encountered a setback when the first animal we selected was euthanized due to pneumonitis. However, we decided to conduct the procedures on this animal despite the circumstances. Thankfully, all subsequent animals that underwent the thoracic procedures remained alive, and we did not encounter any morbidity or mortality in them. We realized that the text in the manuscript may have been unclear and have revised it accordingly.

As well please cite Alaa Obeida, Basma Magdy, Hebatallah Taher, Mohamad Qinawy, Mohamed Elbarbary, Absent Azygos Vein in a neonate with oesophageal atresia and tracheoesophageal fistula. Journal of Pediatric Surgery Case Reports, Volume 90, 2023, 102588, ISSN 2213-5766,  https://doi.org/10.1016/j.epsc.2023.102588.

Thank you very much for this comment. We would like to cite this paper but it does not have an connection to robotic pediatric surgery.

One other important procedure or area which was not mentioned was pelvic operation, given the weight of the piglets it’s ideal of infants in first 6 months of life the time during which we do carry out several pelvic procedures. The pelvic hiatus is very challenging to access open approachand MIS is proven to be better however  in such narrow spaces ergonomics can be challenging especially in those children whose pelvis is anatomically distorted and become smaller than usual such as patient with bladder exstrophy and cloaca patients.

Thank you for your valuable suggestion. Simulation of infant pelvic procedures is crucial for pediatric robotic surgery, especially in cases like Hirschsprung's disease, anorectal malformations, and cloaca. However, it's worth noting that piglet pelvic anatomy differs somewhat from that of human infants, with a narrower, more V-shaped pelvis and a mobile bladder located pseudo-intraperitoneally. In this study, we performed rectal dissections down to the pelvic floor and mobilized the uterus and bladder, but did not evaluate these procedures and thus did not include them in our report. We have revised the manuscript accordingly.

Please cite: Taher HMA, Fares A, Wishahy AMK. Laparoscopic Resurrection of an Old Technique: A New Approach for Total Urogenital Separation and Rectal Pull-Through in Patients with Long Common Channel Cloacal Malformation. J Endourol. 2022 Sep;36(9):1177-1182. doi: 10.1089/end.2021.0724. Epub 2022 Mar 7. PMID: 35152733; PMCID: PMC9422784.; Taher H, Elboraie A, Fares A, Tawfiq S, Elbarbary M, Abdullateef KS. Laparoscopic inguinal hernia repair in bladder exstrophy, a new modified solution to an old problem: A cohort study. Int J Surg Case Rep. 2022 Jun;95:107252. doi: 10.1016/j.ijscr.2022.107252. Epub 2022 May 25. PMID: 35636219; PMCID: PMC9157442.

Thank you very much for your comment, we included PMID: 35152733 in our manuscript.

With remarkable advances of computer vision haptic feedback is diminishing in significance, 3D stereotypic vision is available in your new system which is good what kind of computer vision options are available and can be potentially available.

Thank you very much for your comment. The Versius system offers a basic augmented vision with display of the instruments, their rotation and of the applied monopolar or bipolar cautery as seen in figure 1, 2 and 3. The potential lies in augmented reality with synchronous display of preoperative imaging or artificial intelligence generated zones of high risk of complications being displayed into the field of view of the surgeon.

I would suggest limitations of the study or suggestions for future improvements would be (rather than comparative studies) operating on 2 kg animals, thoracicscopic procedure on a live animal once this is carried out and proven successful the next step would be comparative studies.

Thank you very much for this very important suggestion. We totally agree with you that robotic procedures should also be evaluated in animal models of less than 5kg down to 1500 grams or less. We are currently working on a neonatal piglet model for esophageal surgery, once established robotic procedures can be evaluated with this model, too. We have added this passage to the revises manuscript. Thoracic procedures have been carried out in live animals as well, see our reply above.

Reviewer 2 Report

I rad the manuscript titled: “Evaluation of the Versius robotic system for infant surgery – a 2 study in piglets of less than 10kg body weight: with great interest.

Here are my questions and comments:

1.       Methods section, last paragraph: Evaluation of the surgical procedures: The abbreviations: DeltaRV, etc need to be fully written individually to clarify for the readers for future reference

2.       The abbreviations mentioned in table 1 need to be explained below the table

3.       The setup of the robotic system needs to be explained in detail: monitors, console, arm, boom etc

4.       Were there any data collected for refinement of movements, elimination of surgeon had tremors? They would be really helpful to assess feasibility of the system for live surgery

5.       Is there a unique feature of the Versius system which may be advantageous compared to the contemporary robotics systems?

6.       Is the manufacturer planning to develop suction and stapling devices that could be applied through robotic arms?

7.       Using the system in pediatric patients will require extreme accuracy for insertion of the instruments and memory. How was this assessment made?

8.       What is the longevity of the instruments and arms for the system?

Author Response

please see attachment and here:

Reviewer 2

I read the manuscript titled: “Evaluation of the Versius robotic system for infant surgery – a 2 study in piglets of less than 10kg body weight: with great interest.

Thank you very much!

Here are my questions and comments:

1. Methods section, last paragraph: Evaluation of the surgical procedures: The abbreviations: DeltaRV, etc need to be fully written individually to clarify for the readers for future reference

We corrected the abbreviations in the revised manuscript.

2. The abbreviations mentioned in table 1 need to be explained below the table

We explained the abbreviations and corrected them.

3. The setup of the robotic system needs to be explained in detail: monitors, console, arm, boom etc

Thank you very much: We expanded the description of the robotic system in the materials and methods section to: The setup (Figure 1) consisted of three to four separate arms, each on its own mobile base with an average weight of 100kg. The camera (a 10 mm 3D with either a 0° or 30° scope) was attached to a separate arm. The master console consists of a HD 3D Monitor used with passive 3D glasses. The console grips are a mix between forceps like handpieces for the index finger and thumb and pistol grips. The instruments, consisting of passive, bipolar or monopolar graspers, scissors and hooks were applied as judged by the operating surgeons. The system was covered in sterile drapes and connected to a video system with an additional external monitor and recording capabilities (Sony, Germany). CO2-Insufflation was applied with the Endoflator® (Karl Storz, Tuttlingen), suction and irrigation were generated with the Unimat30® (Karl Storz, Tuttlingen). The sutures used were documented in the separate procedures. All sutures were tied with intracorporal knots, as we deemed the robotic system being especially beneficial in laparoscopic intracorporal knot tying.

4. Were there any data collected for refinement of movements, elimination of surgeon had tremors? They would be really helpful to assess feasibility of the system for live surgery

Thank you for your valuable comment. Our primary objective was to evaluate the feasibility and technical constraints of this system during live surgery. Consequently, we did not gather any data on refining movements, minimizing tremors, or adjusting movement scaling. We concur that assessing these options during live surgery would be beneficial and should be included in future studies.

5. Is there a unique feature of the Versius system which may be advantageous compared to the contemporary robotics systems?

Thank you for your inquiry. Considering the multitude of surgical robotic systems available, numerous platforms share common functionalities, such as wristed instruments equipped with seven degrees of freedom, high-resolution 3D visualization, and optimized ergonomics. Among these features, the Versius system, along with the Senhance system, enables instruments to be directly inserted into the patient without any ports. This is possible due to the automated calculation of the pivot or fulcrum point by the system, resulting in enhanced flexibility compared to robotic systems that require port docking for surgical access. We added this to the discussion section.

6. Is the manufacturer planning to develop suction and stapling devices that could be applied through robotic arms?

We appreciate your inquiry. Despite regular discussions and conferences with the developer throughout our study, we did not engage in deliberations pertaining to the implementation of suction, stapling, or other related devices. Therefore, we cannot provide a detailed assessment of CMR's product line in this regard.

7. Using the system in pediatric patients will require extreme accuracy for insertion of the instruments and memory. How was this assessment made?

Thank you for your important inquiry. The insertion of surgical instruments, as well as the accurate calibration of the pivot or fulcrum point, and memory of the instrument's depth must be performed with utmost precision and safety. This aspect is particularly crucial when operating on children, as the abdominal wall of adults is less compliant and more rigid compared to children and infants. Therefore, safety, accuracy, and precision of port insertion, instrument insertion, and memory of instrument positioning during exchanges were primary goals of our animal study. To achieve this, we employed a specific setup with a three-port approach and varied insufflation pressures ranging from 2-16 mmHg, along with different instruments and cameras to evaluate the calibration of the fulcrum point for laparoscopic and thoracoscopic procedures in the animals. Any collision or bending of instruments, as well as excessive depression with deformation of the body wall, were carefully recorded to evaluate the safety and accuracy of the procedures.

8. What is the longevity of the instruments and arms for the system?

The instruments can be used between 15 and 20 cycles, the arms without any time constraints.

Reviewer 3 Report

I would like to congratulate the authors on their fascinating work regarding this interesting article on the Evaluation of the Versius robotic system for infant surgery.

Despite the major advances in infant surgery, there are still numerous unanswered questions regarding which type of operation has the best results. The manuscript is well-written and the incorporated table and figures make the study easy to follow.

I strongly recommend acceptance for publication of the paper after minor revision:

1) "In the last few years, technological developments in the surgical field have been rapid and are continuously evolving. One of the most revolutionizing breakthroughs was the introduction of the IoT concept within the surgical practice."

I would suggest adding this information in the introduction or discussion section and consider citing the recently published article:

https://pubmed.ncbi.nlm.nih.gov/35746359/

In addition, I would suggest a brief discussion on the Internet of Surgical Things in Telesurgery and Telementoring, while performing infant surgeries.

Author Response

Please see atachment and here

Reviewer 3

I would like to congratulate the authors on their fascinating work regarding this interesting article on the Evaluation of the Versius robotic system for infant surgery.

Thank you very much.

Despite the major advances in infant surgery, there are still numerous unanswered questions regarding which type of operation has the best results. The manuscript is well-written and the incorporated table and figures make the study easy to follow.

Thank you very much, we appreciate your comment.

I strongly recommend acceptance for publication of the paper after minor revision:
1) "In the last few years, technological developments in the surgical field have been rapid and are continuously evolving. One of the most revolutionizing breakthroughs was the introduction of the IoT concept within the surgical practice." I would suggest adding this information in the introduction or discussion section and consider citing the recently published article: https://pubmed.ncbi.nlm.nih.gov/35746359/

Thank you for your insightful comment. We concur with your assessment that the integration of the Internet of Things (IoT) in the realm of surgery presents an opportunity for enhancing the operating theater's functionality through improved communication and connectivity among various surgical devices, image-guided surgery, imaging capabilities, and bolstering patient safety. Your suggestions have been duly noted and will be incorporated into the ongoing discussion.

In addition, I would suggest a brief discussion on the Internet of Surgical Things in Telesurgery and Telementoring, while performing infant surgeries.

Thank you for your valuable suggestion. We have incorporated the following paragraph into the discussion section of our manuscript: " The emergence of digital surgery and the Internet of Surgical Things, including robotic-assisted surgery, has paved the way for a more collaborative approach to surgery, enabling long-distance telesurgery and telementoring. The integration of interconnected robotic-assisted surgical devices, augmented reality, artificial intelligence, image-guided surgery, and autonomous interventions promises to enhance surgical workflow in the operating theater and ultimately bolstering patient safety."

Reviewer 4 Report

Thank you for the opportunity to analyze your interesting article.             

In this article, authors have analyzed the feasibility of Versius® CMR Surgical Robotic system in an “in vivo infant model” with a piglet. Research in medical and surgical pediatrics is a poor relation of medicine, and it is very interesting to see that early in the development of robotics, pediatric surgery is studied.

            Concerning the introduction:

            The introduction is well written, with known data about advantages of minimally invasive surgery and robotic approaches.

            Concerning the methodology:

            The animal model: Piglet seems to be representative of an infant. Legal and ethics information’s are mentioned. 

            The anesthesia protocol is well described.

            The surgical approach and procedures:

-       It’s well described with main informations about modifications required in the piglet model, as it should e with an child: Difficulties with a child for the calculation of the pivot point, and the modification allowing the instruments insertion in the cavity in 2 cm and not 5 cm.

-       Most common procedures were performed, which is a nice performance for all the team.

-       Recorded data are well described

-       Need to have the same abbreviations between recorded data ΔLC, ΔRC, ΔLC, ΔALC and ΔARC and abbreviations of the Table 1 ΔOR, ΔOL, ΔLR

            Concerning the results

            Results are well reported and clearly presented. 

            For the Table 1, need to have same abbreviations in the legend and in the table.

            Surgical procedures are well described. Maybe a small drawn with the robotic setting of all procedure could be interesting with sizes data for example. Or more pictures, 2 is not enough) and some short movies are welcome, unless "industrial secrecy" does not allow it. No major concerns about visceral and digestive procedures. For thoracic procedures, why not doing left upper lobectomy because anatomy is very different with human in the piglet right side. 

            Concerning the discussion:

It’s a well written discussion, with the major results highlighted, and well documented with good references. 

This article is a report of “feasibility” in an animal model. This is a mandatory step to go further. With experienced surgeons and also experienced teams, the Versius® CMR seems to be an interesting robotic system for pediatric surgery, with its “smaller arms” but as authors mentioned, the camera is “too big” 10mm compared to the 3mm used in video assisted procedures. Some works need to be done again to have a “pivot point calculation” child compatible due to the abdominal wall elasticity, and the instruments should only be inserted of 2 cm and not 5cm as in adults. Strenghts and limits of the Versius® CMR are well described without conflict of interest, and the comparison with the the da Vinci® or Senhance® system seems to be objectives. 

Concerning the conclusion:

            No major concerns.

            It’s a well written, easy reading and very interesting article, that just need minor corrections and maybe more illustrations. 

Congratulation to authors for this “first step” and I’m looking forward to the next part with the comparison with VATS and open surgery. 

Author Response

See attachment and here:

Reviewer 4

Thank you for the opportunity to analyze your interesting article.In this article, authors have analyzed the feasibility of Versius® CMR Surgical Robotic system in an “in vivo infant model” with a piglet. Research in medical and surgical pediatrics is a poor relation of medicine, and it is very interesting to see that early in the development of robotics, pediatric surgery is studied.

Thank you very much for your comment.

Concerning the introduction: The introduction is well written, with known data about advantages of minimally invasive surgery and robotic approaches.

Thank you very much.

Concerning the methodology: The animal model: Piglet seems to be representative of an infant. Legal and ethics information’s are mentioned. The anesthesia protocol is well described. The surgical approach and procedures: -It’s well described with main informations about modifications required in the piglet model, as it should be with an child: Difficulties with a child for the calculation of the pivot point, and the modification allowing the instruments insertion in the cavity in 2 cm and not 5 cm. Most common procedures were performed, which is a nice performance for all the team. Recorded data are well described. Need to have the same abbreviations between recorded data ΔLC, ΔRC, ΔLC, ΔALC and ΔARC and abbreviations of the Table 1 ΔOR, ΔOL, ΔLR

Thank you for bringing this to our attention. We have taken note of the error and have corrected the abbreviations in both the table and the text, ensuring that they are consistent and accurate.

Concerning the results: Results are well reported and clearly presented. For the Table 1, need to have same abbreviations in the legend and in the table. Surgical procedures are well described. Maybe a small drawn with the robotic setting of all procedure could be interesting with sizes data for example. Or more pictures, 2 is not enough and some short movies are welcome, unless "industrial secrecy" does not allow it. No major concerns about visceral and digestive procedures. For thoracic procedures, why not doing left upper lobectomy because anatomy is very different with human in the piglet right side.

Thank you for your valuable suggestions. We have updated the abbreviations in the table to ensure consistency and accuracy.

Regarding the surgical setting of the robotic arms, it should be noted that the setup was individualized for each procedure, resulting in a unique configuration. For this reason, we did not provide drawings of the setup. We appreciate your suggestion that performing a left upper lobectomy would likely be a simpler procedure. However, in this particular study, we were performing thoracic procedures on the right side for esophageal surgery. To minimize the potential for stress to the animal, we opted to perform these procedures on one side of the thoracic cavity, rather than performing bilateral thoracoscopic surgery.

Concerning the discussion: It’s a well written discussion, with the major results highlighted, and well documented with good references. This article is a report of “feasibility” in an animal model. This is a mandatory step to go further. With experienced surgeons and also experienced teams, the Versius® CMR seems to be an interesting robotic system for pediatric surgery, with its “smaller arms” but as authors mentioned, the camera is “too big” 10mm compared to the 3mm used in video assisted procedures. Some works need to be done again to have a “pivot point calculation” child compatible due to the abdominal wall elasticity, and the instruments should only be inserted of 2 cm and not 5cm as in adults. Strenghts and limits of the Versius® CMR are well described without conflict of interest, and the comparison with the the da Vinci® or Senhance® system seems to be objectives.

Thank you very much.

Concerning the conclusion: No major concerns.

Thank you very much again.

It’s a well written, easy reading and very interesting article, that just need minor corrections and maybe more illustrations. Congratulation to authors for this “first step” and I’m looking forward to the next part with the comparison with VATS and open surgery.

Thank you very much for your comments. We have added another figure for illustration.

Round 2

Reviewer 2 Report

I thank the authors for taking up on my suggestions. I recommend adding their answer about refining movements and address the questions about instrument memory.

Other than that, I congratulate them for their work. 

Author Response

Reply to the Reviewers

Dear Madam and Sir,

 thank you very much for the opportunity to revise our manuscript. Please find attached our answers and corrections to the reviewers comments.

Ethic approval date

We have included the ethical approval date, which was: October 6th, 2021. This was added in the manuscript.

Reviewer 2:

I thank the authors for taking up on my suggestions. I recommend adding their answer about refining movements and address the questions about instrument memory.

Thank you very much for your recommendation. The refinement of movements through recalibration of the pivot point was found to be unnecessary during the procedures. The system's instrument memory feature allowed for multiple instrument exchanges without requiring recalibration, thereby avoiding any prolongation of the workflow. Interestingly, we discovered that calibration of the pivot point can also be performed 'in free air' by manually holding the instrument at the desired fulcrum point during calibration, providing a workaround for repeated unsuccessful calibration attempts. While we applied this method in our study, it was not found to be necessary for docking the system. However, it does afford the surgeon greater flexibility in determining the optimal placement of the pivot point. We have added this to our manuscript.

Other than that, I congratulate them for their work.

Thank you very much for your supportive comments.

We hope to have answered all your questions. We would be very happy if you accept our revised manuscript for publication in your journal.

Prof. Dr. med. Robert Bergholz 

UKSH Campus Kiel